# Mass Spectrometry as Alternative Method to Identify and Monitor Non-Secretory Progressive Disease in Patients with Multiple Myeloma

**DOI:** 10.3390/biomedicines12061153

**Published:** 2024-05-23

**Authors:** Cristina Agulló, Noemí Puig, Teresa Contreras, Sergio Castro, Borja Puertas, Verónica González-Calle, Beatriz Rey-Búa, María Victoria Mateos

**Affiliations:** 1Clinical Biochemistry Department, University Hospital of Salamanca (HUSAL), 37007 Salamanca, Spain; 2Hematology Department, University Hospital of Salamanca (HUSAL), IBSAL, IBMCC (USAL-CSIC), CIBERONC, 37007 Salamanca, Spain

**Keywords:** multiple myeloma, non-secretory disease, mass spectrometry, treatment monitoring

## Abstract

Introduction: After receiving different lines of treatment, multiple myeloma patients tend to present with less secretory and more frequent extramedullary disease. These features make treatment monitoring and follow-up very complex since they have to be based on the use of imaging methods and/or bone marrow aspirations or biopsies. Objective: To present the case of a patient with myeloma progressing with non-secretory bone disease and to discuss the potential impact of mass spectrometry as a new highly sensitive method able to identify the monoclonal protein (MP) in the serum of these types of patients. Materials and Methods: Informed consent was signed by the patient prior to receiving each line of treatment. The clinical information and images were obtained from anonymized electronic files. The mass spectrometry was performed with the Immunoglobulin Isotypes (GAM) assay for the mass spectrometry EXENT^®^ Analyser Technology from Binding Site, part of Thermofisher. Results: A 73-year-old male with IgG kappa multiple myeloma progressing with a new lytic lesion after receiving 14 cycles of Talquetamab as a third line of therapy who, due to the non-secretory nature of the disease at this point, could not be enrolled in a clinical trial, thus limiting his therapeutic options. The mass spectrometry was able to identify and quantify the presence of the patient’s MP when the serum protein electrophoresis and immunofixation were still negative and therefore could have been used to confirm the progression, to permit the inclusion of the patient in a clinical trial and to further monitor the disease response. Conclusions: The higher sensitivity of the mass spectrometry methods to detect the MP in patients with myeloma and other monoclonal gammopathies translates into better identification of the disease progression, permits the inclusion of more patients in clinical trials and facilitates treatment monitoring.

## 1. Introduction

Multiple myeloma (MM) is characterized by the proliferation of the malignant clonal Plasma Cells (PCs) accumulating in the bone marrow (BM) that produce and secrete a monoclonal immunoglobulin also named M-protein (MP). This MP is often an intact immunoglobulin, but it can also consist of Free Light Chains (FLCs), a combination of both or, more rarely, heavy chains only. The identification and quantification of the MP is key for the diagnosis and monitoring of the disease since its amount is considered a surrogate of the tumor burden. Serum protein electrophoresis (SPEP) and immunofixation (IFE) and the analysis of the FLCs when indicated are the methods recommended by the International Myeloma Working Group (IMWG) to determine the treatment response and monitor for relapse [1]. However, their sensitivity is limited, and the interpretation of electrophoretic results is sometimes very subjective, especially when the patients are engaged in a complete response. The recent improvements in multiple myeloma therapies have led these patients to achieve very high rates of complete response; unfortunately, however, they continue to relapse. This fact demonstrates that the use of the conventional methods to detect lower concentrations of MP is suboptimal and, therefore, leads to the need for more sensitive techniques to identify the tumor burden in the peripheral blood, as achieved with minimal residual disease (MRD) by the next-generation techniques in bone marrow. Also, with the evolution of the disease and after receiving various lines of treatment, MM patients typically develop extramedullary disease, often producing very low levels of serum MP and hemoglobin and significantly higher levels of Lactate Dehydrogenase (LDH) [2,3]. With no detectable disease in the serum or urine with the conventional methods, we need to follow the relapse via imaging techniques regarding the evolution of the lytic lesions. This lack of measurable disease in the serum and urine in patients with MM hampers the treatment monitoring and precludes their inclusion in the majority of clinical trials.

As an alternative to the standard methods, mass spectrometry is a highly sensitive and specific method that is able to detect very low levels of MP. By using the unique molecular mass produced by the malignant clone to accurately identify the presence or absence of the disease during a follow-up, MS is also able to quantify very low concentrations of MP over time in patients with monoclonal gammopathies, a feature that could be especially relevant under the previously mentioned circumstances [3,4].

Mass spectrometry results have demonstrated a significant clinical impact in terms of the median progression-free survival (mPFS) when patients transition from a negative to a positive result during their follow-up. In our clinical case, we highlight the importance of identifying and quantifying very low levels of MP by MS in the progression of extramedullary disease while the MPs in the serum or urine are not measurable and undetectable by the conventional methods.

## 2. Case Presentation

A 73-year-old male was diagnosed in September 2019 with IgG Kappa MM, R-ISS 1, presenting with a vertebral plasmacytoma and multitopic bone disease (a lytic lesion with a soft tissue component entering the spinal canal at T6, destruction of T5 and further lytic lesions at C7, L5 and the seventh right posterior costal arch). Further, 22% PCs were identified in the bone marrow aspiration, and no translocations of the IGH gene or deletion of the P53 gene were found by FISH in isolated PCs. 

He received first-line treatment with five cycles of VTD (bortezomib, thalidomide and dexamethasone) starting in September 2019, after which thalidomide was switched to lenalidomide due to painful peripheral neuropathy. Due to the COVID-19 pandemic, the decision was made to delay the initially planned autologous stem-cell transplantation (ASCT) and to administer him three further cycles of VRD, after which he achieved a complete response (CR) with negative Measurable Residual Disease (MRD) in the bone marrow by Next-Generation Flow. Then, he received high-dose chemotherapy with melphalan 200 mg/m^2^ followed by ASCT on 6 June 2020, maintaining the MRD-negative status 3 months afterwards. Maintenance treatment with a standard dose of lenalidomide was started in September 2020, which was reduced to 5 mg in December of 2020 due to grade 3 skin toxicity. In July 2021, still under lenalidomide treatment, he progressed with multiple new lytic lesions identified in a low-dose CT scan. 

He was then started on DVd (daratumumab, bortezomib and dexamethasone) as a second-line treatment in August 2021, receiving a total of eight cycles until March 2022 and achieving stable disease as the best response. In April 2022, during an in-clinic observation, he complained of asthenia and pain in the distal femur. The physical examination was unremarkable, but a PET/CT was requested, which was highly suggestive of disease progression with new medullary and extramedullary involvement. At this time point, the patient presented with 7 g/L of MP detected by SPEP, 72.39 mg/L of Serum Free Light Chain (sFLC) kappa, 3.18 mg/L of sFLC lambda, an sFLC ratio of 22.76 and a kappa Bence Jones protein of 0.01 g/24 h. Using the Immunoglobulin Isotypes (GAM) assay for the mass spectrometry (MS) EXENT^®^ Analyser in the same sample, a non-glycosilated IgG Kappa MP was identified with an *m*/*z* of 11,796 and a concentration of 6.95 g/L (Figure 1a). 

At this moment, the patient was offered (and accepted) to be enrolled in a clinical trial involving the use of single-agent Talquetamab, a bispecific antibody against CD3 and GPRC5D that redirects T cells to mediate the killing of GPRC5D-expressing myeloma cells. By the end of January 2023, after receiving eight cycles of Talquetamab, he achieved CR with no detectable residual disease in the bone marrow by Next-Generation Flow, and the examination by PET/TC showed a complete metabolic response in all the hypermetabolic lesions previously identified. Unfortunately, at the beginning of July 2023, before starting the 15th cycle of treatment, the patient complained again of bone pain in the right shoulder area. A follow-up PET/CT was requested, which showed a hypermetabolic lytic lesion (SUVmax = 7.7) located in the medial third of the right clavicle that was not presented previously and thus suggestive of progressive disease (Figure 2). At this point, no measurable disease was detectable in the serum or urine by SPEP/IFE or mass spectrometry.

In summary, our patient developed progressive disease after receiving 14 cycles of Talquetamab based on the appearance of new hypermetabolic lytic lesions in a PET-CT scan, but no MP was detectable at this time point in the serum or urine using the conventional methods. By the end of July of 2023, with the SPEP/IFE still negative in the serum and urine, mass spectrometry was already able to identify the reappearance of the original patient’s MP by identifying a small kappa peak with the exact *m*/*z* identified at relapse (Figure 1b).

On the subsequent follow-up samples, corresponding to October, November and December of 2023 (Figure 3), mass spectrometry confirmed the presence of the patient MP, which was also quantifiable and was actually progressively increasing in size, being 0.399 g/L in December of 2023. Most importantly, at this point in time, the MP remained unde tectable using SPEP and IFE both in the serum and urine. 

Furthermore, in November, a follow-up PET/CT shows the metabolic progression of the same lytic lesion since the relapse with SUVmax = 19, currently with the associated pathological fracture supporting the progressive myeloproliferative disease (Figure 4).

## 3. Discussion

We present the case of a patient with MM relapsing after three lines of therapy due to progressive bone disease in whom the lack of serum/urine-measurable disease hampers the patient’s monitoring and precludes his inclusion in a clinical trial. A lack of measurable disease at diagnosis occurs in a low proportion of MM patients but increases upon subsequent relapses. These patients need to be followed with bone marrow aspirations or imaging methods, making the monitoring painful and cumbersome [5,6]. Importantly, the majority of clinical trials include the presence of measurable disease in the serum or urine as mandatory inclusion criteria. In the case presented herein, we decided to adopt a very careful watch and wait approach, hoping that the patient would soon develop measurable disease in the serum, urine or by imaging since we believe that he would benefit most from being enrolled in a clinical trial involving the use of CAR-T cells or a non-GPRC5D-based bispecific monoclonal antibody. So, currently, the therapeutical options for this patient are limited.

The mass spectrometry was indicative of the progression in the serum very soon after the PET/CT because the appearance of the same MPs identified at relapse, with the same molecular mass, was identified in the serum sample, highlighting the high sensitivity and specificity compared to the conventional techniques. Also, the mass spectrometry results coincide with the imaging techniques, which supports that the patient is in relapse. Furthermore, according to previous studies, a loss of mass spectrometry negativity is associated with the disease progression and an inferior PFS [4]. Therefore, we believe that the MS introduction in the clinical routine will help to confirm the earlier progression, which might enable not only the inclusion of patients with these characteristics in future clinical trials but also the easy and non-invasive monitoring method to evaluate the treatment response. Further investigations are needed to fully establish how early MS can identify the disease when compared to the conventional techniques. This understanding will enable not only the adequate use of expensive and invasive techniques such as imaging and MRD, respectively, but also to correctly establish the therapeutic approaches in patients with extramedullary disease.

## 4. Conclusions

The higher sensitivity of mass spectrometry methods as compared to SPEP/IFE to detect the MP in patients with MM and other monoclonal gammopathies can translate into the earlier identification of disease progression and might facilitate in the near future the adoption of therapeutic options, such as patient inclusion in clinical trials and changes in the treatment strategy. Importantly, MS will also enable avoiding the need to perform invasive bone marrow techniques or imaging methods that are expensive and inconvenient for the patients. 

## Figures and Tables

**Figure 1 biomedicines-12-01153-f001:**
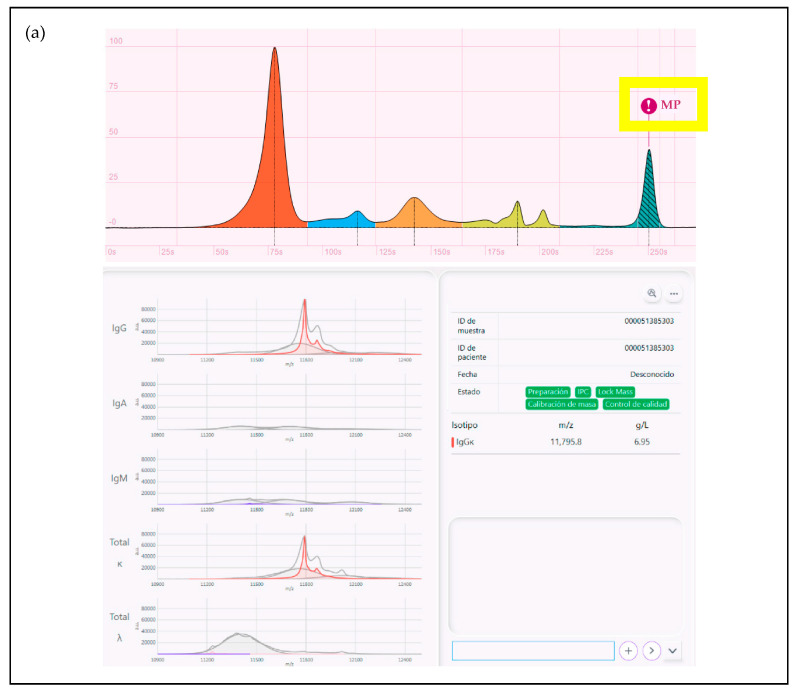
(**a**) SPEP (positive, showing the presence of MP IgG kappa migrating in the gamma region of proteinogram, with 7 g/L concentration) and mass spectra (positive, showing the presence of MP IgG kappa with 11,796 m/z and 6.95 g/L concentration) regarding patient relapse in 2022; (**b**) IFE (negative) and mass spectra (positive, showing the presence of a small kappa peak with the exact m/z identified previously, 11,796 m/z).

**Figure 2 biomedicines-12-01153-f002:**
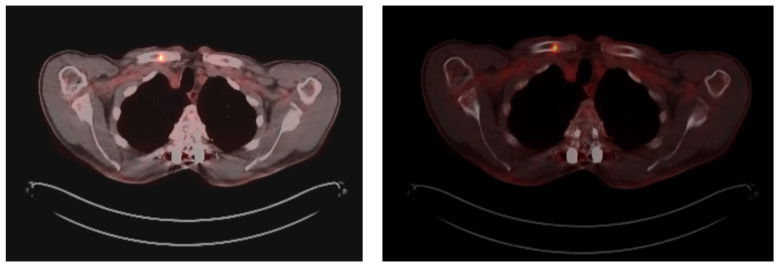
Hypermetabolic lytic lesion (SUVmax = 7.7) in the medial third of the right clavicle, not present in the previous study and highly suggestive of progressive disease.

**Figure 3 biomedicines-12-01153-f003:**
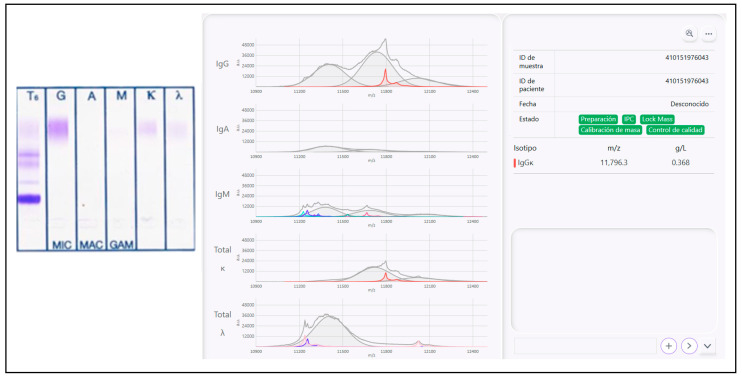
IFE (negative) and mass spectra (positive, showing the presence and concentration of an IgG kappa peak with the exact m/z identified at relapse, 11,796 m/z, with 0.368 g/L of MP concentration).

**Figure 4 biomedicines-12-01153-f004:**
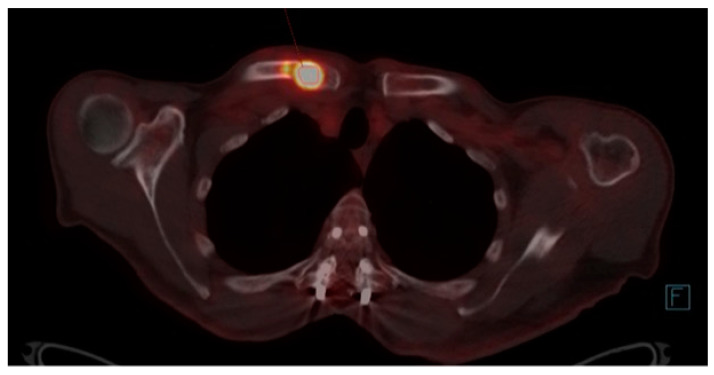
Progression of hypermetabolic lytic lesion (SUVmax = 19) in the medial third of the right clavicle four months after the previous one.

## Data Availability

Data is contained within the article. More data are not available for patient privacy reasons.

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
