# Peer review of "Mass Spectrometry as Alternative Method to Identify and Monitor Non-Secretory Progressive Disease in Patients with Multiple Myeloma"

_biomedicines, 2024, doi:10.3390/biomedicines12061153_

Round 1

Reviewer 1 Report

Comments and Suggestions for Authors

The authors provide a case report focusing the possible role of mass spectrometry  to identify non-Secretory progressive disease. 

The paper is clear and well written. 

My concern is about the clinical applicability and costs of using this specific methods as routine test. Although the high sensitivity, it should be less applicable for small laboratory 

Comments on the Quality of English Language

The english is good

Author Response

Thank you so much for your comment.  This methodology is, for the moment, an emergent technique, although a similar methodology is already used in a daily practical routine in the Mayo clinic where it has substituted the conventional techniques, SPEP and IFE. Mass spectrometry is the future trend to measure MP and is very possible that soon treatment response criteria from the International Myeloma Working Group will be updated with MS.

Even if MS is more expensive than the current electrophoretic methods, this technique is less expensive than the invasive techniques that the hematologists use to determine the minimal residual disease in Bone Marrow with the important advantage that can be performed in peripheral blood and by consequence, less invasive and painful for the patient.

Reviewer 2 Report

Comments and Suggestions for Authors

The authors present a case report in which a patient with non-secretory multiple myeloma was able to be monitored for disease using mass spectrometry.

A few comments/questions:

In the abstract, the conclusion is too strong. It cannot be determined that the method "translates into a better identification of disease progression, permits the inclusion of more patients in clinical trials and facilitates treatment monitoring" from data from only one patient.

 In the figures 1b, left, and figure 3, left, there are bands under kappa and lambda. Could those bands not be used to determine disease abundance?

I am curious as to how the MPs can be quantified using mass spectrometry.

What if there were a new clone rather than the same clone as before? Could you still use mass spectrometry?

Figure 1a -- What does "CM" mean?

Comments on the Quality of English Language

The text is very readable, although there are a few errors, primarily words that shouldn't be capitalized, as well as errors in the Reference list.

Author Response

Thank you so much for your comments, and I’m going to answer each of them below:

  1. we have modified the conclusion in based your comment.

“The higher sensitivity of mass-spec methods as compared to SPEP/IFE to detect the MP in patients with MM and other monoclonal gammopathies can translate into earlier identification of disease progression and might facilitate in near future the adoption of therapeutic options, such the patient inclusion clinical trials and change of treatment strategy. Importantly MS will also allow to avoid the need to perform invasive bone marrow techniques or imaging methods, that are expensive and inconvenient for the patients.”

  1. These bands, identified by IFE, are typical oligoclonal bands and the migration of these thin bands does not coincide with the original migration. His monoclonal protein migrates at the end of the electrophoresis. So these bands have no information about his disease.

  1. The monoclonal protein peaks are quantified indirectly in combination with immunoturbidimetry. Quantitation of monoclonal protein peaks detected by the EXENT Analyser is performed automatically. The algorithm for this equates the modelled total light chain peak area detected for either IgG, IA or IgM using EXENT assay to the total IgG, IgA or IgM concentration by Optilite. It the expresses the modelled monoclonal peak area as a fraction of that concentration.
  2. Yes, the MP is detected based on the specific molecular mass of the MP produced by the malignant clones. If more clones existed or surged after treatment these clones would present a different m/z and by consequence would be identified as a different MP coming from a different malignant clone.

  1. Sorry, in English we call "Componente Monoclonal" the monoclonal peak by electrophoresis, and the software calls the monoclonal protein with this abbreviation "CM". Already modified.
